# Microbiota composition and mucosal immunity in patients with asymptomatic diverticulosis and controls

**Tessel M. van Rossen**[1][ORCID]*, **Rogier E. Ooijevaar**[2][ORCID], **Johan Ph. Kuyvenhoven**[3], **Anat Eck**[1], **Herman Bril**[4], **René Buijsman**[5], **Marja A. Boermeester**[6], **Hein B. A. C. Stockmann**[7], **Niels de Korte**[7‡], **Andries E. Budding**[8‡]

1 Department of Medical Microbiology & Infection Control, Amsterdam Institute for Infection & Immunity, Amsterdam UMC, location VUmc, Vrije Universiteit Amsterdam, Amsterdam, The Netherlands, 2 Department of Gastroenterology & Hepatology, Amsterdam Gastroenterology Endocrinology Metabolism, Amsterdam UMC, location VUmc, Vrije Universiteit Amsterdam, Amsterdam, The Netherlands, 3 Department of Gastroenterology & Hepatology, Spaarne Gasthuis, Hoofddorp, The Netherlands, 4 Department of Pathology, Spaarne Gasthuis, Hoofddorp, The Netherlands, 5 Department of Traumasurgery, Amsterdam UMC, location AMC, Amsterdam, The Netherlands, 6 Department of Surgery, Amsterdam Gastroenterology Endocrinology Metabolism, Amsterdam UMC, location AMC, University of Amsterdam, Amsterdam, The Netherlands, 7 Department of Surgery, Spaarne Gasthuis, Hoofddorp, The Netherlands, 8 inBiome B.V., Amsterdam, The Netherlands

☯ These authors contributed equally to this work.
‡ NK and AEB authors contributed equally and should be considered last author.
* t.vanrossen@amsterdammumc.nl

**Data Availability Statement:** All relevant data are within the manuscript and its Supporting Information files.

## Abstract

### Introduction

The etiology of diverticulosis is still poorly understood. However, in patients with diverticulitis, markers of mucosal inflammation and microbiota alterations have been found. The aim of this study was to evaluate potential differences of the gut microbiota composition and mucosal immunity between patients with asymptomatic diverticulosis and controls.

### Methods

We performed a prospective study on patients who underwent routine colonoscopy for causes not related to diverticular disease or inflammatory bowel disease. Participants were grouped based on the presence or absence of diverticula. Mucosal biopsies were obtained from the sigmoid and transverse colon. Microbiota composition was analyzed with IS-pro, a 16S-23S based bacterial profiling technique. To predict if patients belonged to the asymptomatic diverticulosis or control group a partial least squares discriminant analysis (PLS-DA) regression model was used. Inflammation was assessed by neutrophil and lymphocyte counts within the taken biopsies.

### Results

Forty-three patients were enrolled. Intestinal microbiota profiles were highly similar within individuals for all phyla. Between individuals, microbiota profiles differed substantially but regardless of the presence (n = 19) of absence (n = 24) of diverticula. Microbiota diversity in both sigmoid and

**Funding:** Anat Eck was supported by The Netherlands Organisation for Health Research and Development (ZonMw), grant number 95103009. inBiome B.V. and ZonMw supported in the form of salaries for authors [AEB, AE], but did not have any additional role in the study design, data collection and analysis, decision to publish, or preparation of the manuscript. The specific roles of these authors are articulated in the 'author contributions' section.

**Competing interests:** I have read the journal's policy and the authors of this manuscript have the following competing interests: AEB reports personal fees of inBiome B.V. (employee/stock holder). Anat Eck was supported by The Netherlands Organisation for Health Research and Development (ZonMw), grant number 95103009. This does not alter our adherence to PLOS ONE policies on sharing data and materials.

transverse colon was similar in all participants. We were not able to differentiate between diverticulosis patients and controls with a PLS-DA model. Mucosal lymphocyte counts were comparable among both groups; no neutrophils were detected in any of the studied biopsies.

## Conclusions

Microbiota composition and inflammatory markers were comparable among asymptomatic diverticulosis patients and controls. This suggests that the gut microbiota and mucosal inflammation do not play a major role in the pathogenesis of diverticula formation.

## Introduction

Diverticulosis is a common gastrointestinal disorder that is usually asymptomatic (asymptomatic diverticulosis; AD). Since it is often found incidentally during routine endoscopic or radiological exam, a true incidence and prevalence of diverticulosis is lacking. It is estimated that roughly 67% of the adult population will develop diverticulosis during their lifespan [1]. Several risk factors have been proposed such as a low fiber diet, smoking, obesity, male sex, aging, and lack of physical exercise [2]. The etiology of diverticulosis is still poorly understood. Genetics and increased colonic intraluminal pressure appear to play an important role [3]. Given the gut microbiota composition is influenced by the aforementioned risk factors, the gut microbiota could be implicated in the pathophysiology of diverticulosis.

Diverticulitis or diverticular disease occurs when a diverticulum becomes inflamed. Clinical symptoms such as abdominal tenderness, constipation or diarrhea, fever and pain can occur. It is still poorly understood what drives a diverticulum to become inflamed. A common but unproven hypothesis states that obstruction and microtrauma in a diverticulum, leads to perforation and subsequent infection. Based on this hypothesis the use of antibiotics for diverticulitis is common practice [2]. However, a recent study suggests that antibiotic treatment may not improve the outcome in diverticulitis [4].

The gut microbiota composition and function has been implicated in health and several diseases, such as *Clostridioides difficile* infection, inflammatory bowel disease, and irritable bowel syndrome [5]. Several studies have shown a different composition in gut microbiota compared to controls in (uncomplicated) diverticulitis [1, 6]. However, there is a paucity of microbiota data in patients suffering from AD. The gut microbiota could be implicated in the pathophysiology of diverticulosis and subsequent diverticulitis. With the recent advances in molecular-techniques it is now possible to map the microbiota from different bodily surfaces such as the bowel mucosa. The interspace region (IS)-pro technique, a 16S-23S based bacterial profiling technique is validated for use in the intestinal microbiota [7].

The etiology of diverticulosis and subsequent diverticulitis may very well be multifactorial including changes in microbiota composition and function. The aim of this study is to characterize mucosal inflammation and differences in colonic mucosal microbiota in individuals with diverticulosis as compared to a control population. This can contribute to further unraveling the etiology of diverticulosis and guide future research on treatment and prevention.

## Materials and methods

### Study design

A prospective single-center study was performed in patients undergoing routine follow-up colonoscopy for causes not related to diverticulitis or inflammatory bowel disease. The study

was conducted on a convenience sample size. All patients underwent routine bowel lavage for colonoscopy. When patients enrolled into our study, colonoscopy was performed by a single gastroenterologist. Participants were enrolled into either the asymptomatic diverticulosis group (AD group) or the no diverticulosis group (control group), based on the findings during colonoscopy. When diverticula were encountered during colonoscopy five biopsies were taken from the mucosa surrounding the diverticula in the sigmoid colon and five biopsies were taken from the transverse colon as control biopsies. When no diverticula were encountered during colonoscopy, five biopsies were obtained from the sigmoid and transverse colon at random.

## Patients

Patients aged 18 years or older undergoing a routine colonoscopy were eligible for recruitment. Exclusion criteria were: suspicion of diverticular related complaints, proven history of symptomatic diverticulitis, history of inflammatory bowel disease or the use of anticoagulants or platelet aggregate inhibitors unless stopped one week prior to colonoscopy. Written informed consent was obtained from all subjects prior to participation. Study approval was provided by the local research ethics committee, *Medisch-Ethische Toetsingscommissie (METC) Noord-Holland*. All research was performed in accordance with relevant guidelines and regulations.

## DNA isolation and amplification

After harvesting the colonic biopsies were washed in phosphate buffered saline to remove residual fecal material and non-adherent bacteria. Subsequently, biopsies were placed in Eppendorf tubes and snap frozen in liquid nitrogen and stored at -20˚C. After thawing of the samples, one ml of NucliSENS lysisbuffer, containing guanidine thiocyanate, was added to each Eppendorf tube and shaken at 1400rpm (Thermomixer comfort, Eppendorf, Hamburg, Germany) for five minutes. Afterwards, all samples were centrifuged for four minutes at 12.000g and added to the easyMag container for total DNA extraction with the NucliSENS easyMag automated DNA isolation machine (Biomérieux, Marcy l'Etoile, France). We have described DNA isolation in detail previously [7].

## Microbiota and data analysis

Amplification of 16S-23S IS-regions was performed with the IS-pro microbiota assay (inBiome B.V., Amsterdam, the Netherlands). IS-profiling was done as described previously [7]. All data were pre-processed with the IS-pro proprietary software suite (inBiome B.V., Amsterdam, the Netherlands). This process resulted in peak profiles with the length of each peak, measured in nucleotides, representing a IS fragment corresponding to a specific bacterial species. The intensity of each peak, measured in relative fluorescence units (RFU), reflected the quantity of PCR product and corresponded to the abundance of that species. Finally, phylum-specific fluorescent labels categorized peaks into three phylogenetic groups: *Proteobacteria*, *Bacteroidetes* or *Firmicutes*/*Actinobacteria*/*Fusobacteria*/*Verrucomicrobia* (FAFV). All intensities were log2 transformed. Log2 transformation of complex profiles compacts the range of variation in peak heights, reducing the dominance of high peaks and including less abundant species of the microbiota in downstream analyses. This results in improved consistency of estimated correlation coefficient, lower impact of inter-run variation and improved detection of less prominent species. This conversion was used in all downstream analyses.

**Cosine distance and diversity analysis.** To analyze similarities between samples, a cosine distance analysis and a diversity analysis was performed. Cosine distance analysis was

performed per phylum and for total microbiota composition at species level within individuals (sigmoid vs transverse colon biopsies) and between individuals (sigmoid vs sigmoid colon) for asymptomatic diverticulosis patients and controls. Diversity was calculated both per phylum and for the overall microbial composition (by pooling all phyla together). Within-sample diversity was calculated as the Shannon index [8]. Dissimilarities between samples, or between-sample diversity, was represented in a dissimilarity matrix that was built using the cosine distance measure [9]. Diversity analysis was performed using the vegan software package in R.

**Partial least squares discriminant analysis (PLS-DA).**   A partial least squares discriminant analysis (PLS-DA) regression model was used for the prediction of clinical status of samples; i.e. whether it belonged to a diverticulosis patient or to a control subject [10]. The PLS-DA model was constructed on the basis of four different datasets: one for each of the three separate phylum groups and one for the overall microbial composition by pooling all phyla. Only the top 25% most variable predictors were considered in the analysis. PLS-DA model validation was carried out by a 10-fold cross validation procedure. The PLS-DA was described *in extenso* in a previous study performed by us [11]. PLS-DA analysis was performed using the DiscriMiner package in R (version 2.15.2). All data visualizations were performed with the Spotfire software 10 package (TIBCO, Palo Alto, CA, USA).

**Clustered heat map.**   For a global analysis of all versus all samples, we generated a clustered heat map. First, a correlation matrix was generated by means of cosine correlation, then clustering was done with the unweighted pair group method with arithmetic mean (UPGMA). The heat map was created with the Spotfire 10 software package (TIBCO, Palo Alto, CA, USA, https://www.tibco.com/products/tibco-spotfire).

## Histology

The presence of inflammatory changes was assessed as previously described by performing a neutrophil and lymphocyte count on 10 different colonic fields at 40 x magnification [12]. Hematoxylin-eosin staining was performed. Lymphocytes were identified with anti-CD3 antibodies (ready to use rabbit anti human polyclonal antibodies, DAKO, Copenhagen, Denmark), for neutrophils anti-CD15 antibodies (ready to use mouse anti human monoclonal antibodies, DAKO, Copenhagen, Denmark) were used. The number of lymphocytes and neutrophils were scored both at the bottom of the crypts and in the crypts as a whole. For histological evaluation the means of lymphocyte and neutrophils infiltrate were compared using the Mann-Whitney test. A P value of <0.05 was considered statistically significant.

## Results

### Patient characteristics

A total of 43 patients were enrolled of which 19 had AD and 24 had no diverticulosis. Patient characteristics and indications for colonoscopy are shown in Table 1. The asymptomatic diverticulosis group consisted of more males than the control group; however, this difference was not statistically significant (74 vs. 38%, p = 0.063). Patients in the AD group were significantly older than patients in the control group (mean age 66.0 vs. 56.4 years, p = <0.001). Six out of 19 AD patients were smokers, compared to 3/24 controls, and alcohol consumption was slightly higher in the AD group compared to the control group. None of the patients received antibiotics via the hospital prior to colonoscopy. The number of patients with specific comorbidities was similar in the two groups, except that gastro-intestinal disorders and malignancies (but not colon cancer) were more frequent in the control group compared to the AD group.

**Table 1. Patient characteristics.** *e.c.i.: e causa ignota.*

| Characteristic | Asymptomatic diverticulosis group | Control group |
|---|---|---|
| | N = 19 | N = 24 |
| Male gender | 14 (74%) | 9 (38%) |
| Age in years | 66.0 (62.7–69.3) | 56.4 (52.7–60.2) |
| Smoking | 6 | 3 |
| Alcohol use (standard volumes/week) | 5.9 (2.1–9.6) | 4.1 (1.4–6.8) |
| Laxative use | 3 | 1 |
| Antibiotic use | 0 | 0 |
| *Comorbidity*: | | |
| Cardiovascular | 9 | 8 |
| Hypertension | 5 | 6 |
| Pulmonary | 1 | 3 |
| Renal | 0 | 0 |
| Gastro-intestinal | 0 | 4 |
| Liver disease | 0 | 1 |
| Malignancy | 2 | 8 |
| Colon cancer | 0 | 0 |
| Rheumatologic | 3 | 1 |
| Neurologic | 7 | 7 |
| Diabetes mellitus | 2 | 4 |
| Immunocompromised | 0 | 0 |
| *Indication for colonoscopy*: | | |
| Hematochezia | 1 | 3 |
| Change in bowel habit | 2 | 2 |
| Anemia e.c.i. | 0 | 1 |
| Obstipation | 0 | 3 |
| Screening and follow up colorectal carcinoma | 2 | 4 |
| Follow up after polypectomy | 13 | 9 |
| Other | 1 | 2 |

Gastro-intestinal comorbidity in the control group concerned esophagitis (n = 2), aspecific colitis (n = 1) and irritable bowel syndrome (n = 1).

## Intestinal microbiota analysis

The *Firmicutes* to *Bacteroidetes* ratio is commonly used to describe the gut microbiota composition. A higher *Firmicutes/Bacteroidetes* ratio is associated with dysbiosis [13, 14]. Therefore, we compared the relative abundance of these phyla between the AD and control group. In the AD group we found a *Firmicutes/Bacteroidetes* ratio of 38%/62% (variance 0.018). Almost identical proportions were found in the control group (*Firmicutes* 37%/*Bacteroidetes* 63%, variance 0.021) with no statistical differences between the two groups (p = 0.69, Students T-test). Also, the total load of bacteria of the *Proteobacteria* phylum was similar between diverticulosis patients and controls (p = 0.56, Students T-test).

 **Cosine distance.** Cosine distance analysis was performed to compare microbiota compositions in participants with AD and controls. To detect potential diverticula-related differences in microbiota composition, we assessed cosine distances of bacterial profiles of the sigmoid and transverse colon within individuals with and without AD. We observed slight location-related differences in both groups–most clearly for the *Proteobacteria* phylum–but the range of

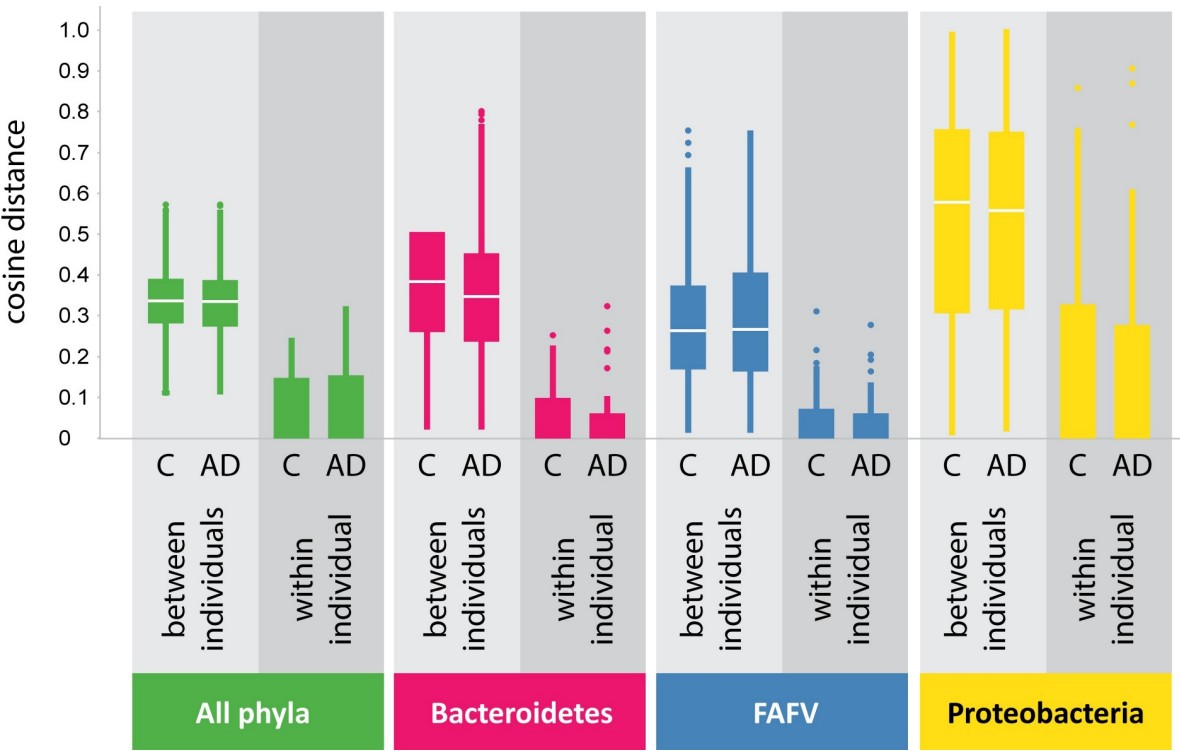

**Fig 1. Cosine distance analysis of intestinal microbiota profiles within and between individuals.** Microbiota profiles are highly similar within individuals for all phyla, regardless of disease status (C = control, AD = asymptomatic diverticulosis). Between individuals microbiota profiles are dissimilar. FAFV: *Firmicutes/Actinobacteria/Fusobacteria/Verrucomicrobia*.

these intra-individual differences was comparable in the AD and control group. Secondly, we assessed if the microbiota composition differed between participants with and without AD. Therefore, we performed an inter-individual comparison of all diverticular biopsies in the AD group and an inter-individual comparison of all sigmoid colon biopsies in the control group (Fig 1). Overall, correlations of microbiota profiles between patients were low. The variation of bacterial profiles between individuals was similar in the AD and control group. If there was a specific diverticulosis signature, higher similarity within the AD group than within the control group would be expected.

**Diversity.** Microbiota diversity of all biopsies was analyzed by calculating the Shannon diversity index (Fig 2). The highest diversity was detected in the *Bacteroidetes* phylum. No differences were observed in the phylum-specific and all phyla pooled diversity indices of transverse and sigmoid colon biopsies. Furthermore, bacterial diversity was similar for individuals with and without AD.

**Partial least squares discriminant analysis (PLS-DA).** A PLS-DA model was constructed to predict whether a biopsy belonged to an individual with AD or a control subject, based on the association between specific bacterial species and the presence of AD. We developed a model that could distinguish between individuals with and without AD with an accuracy of 74%. However, the model showed poor performance when performing 10-fold cross-validation (Fig 3). This suggests that the biopsies did not contain any bacterial species specific for the presence or absence of diverticula.

**Clustered heat map.** We performed a cluster analysis based on microbiota profile similarity between single biopsies. A heat map and dendrogram were constructed including all samples of the AD group and controls (Fig 4). Most resulting 'clusters' consisted of biopsy pairs

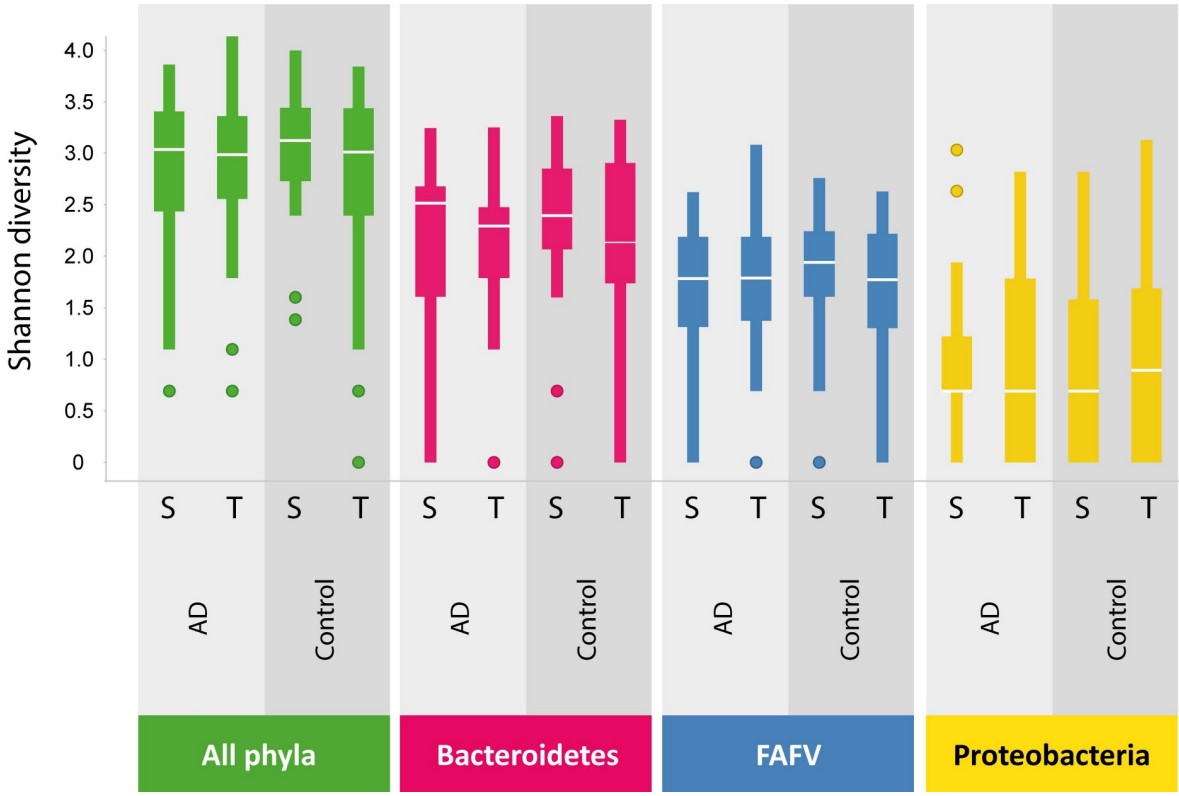

**Fig 2. Diversity analysis of intestinal microbiota per phylum.** Diversity is highest for *Bacteroidetes*, followed by FAFV group and *Proteobacteria*. Diversity is not different in sigmoid (S) and transverse (T) colon or for asymptomatic diverticulosis (AD) patients or controls. FAFV: *Firmicutes/Actinobacteria/Fusobacteria/Verrucomicrobia*.

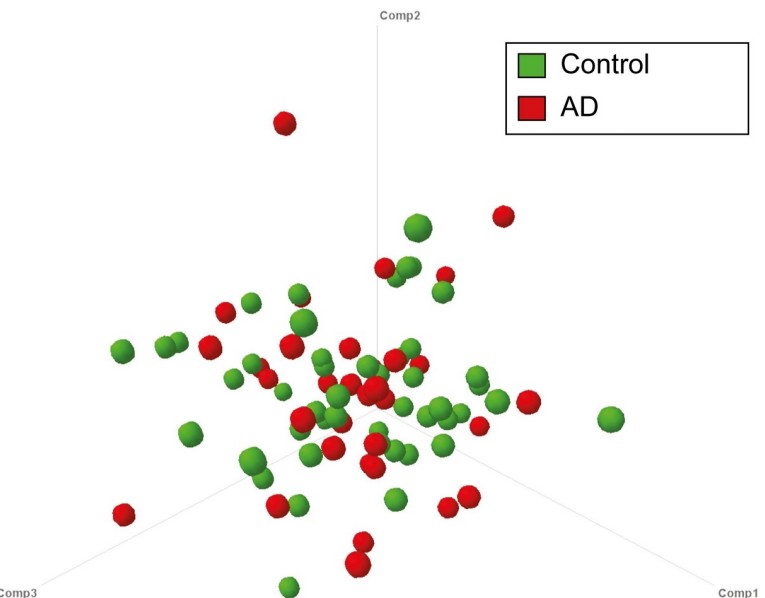

**Fig 3. Three dimensional PLS-DA scores plot of microbiota samples based on the most discriminative PLS components.** It can be clearly seen that there is no separation, suggesting the absence of discriminative species for either state (asymptomatic diverticulosis, AD, or controls).

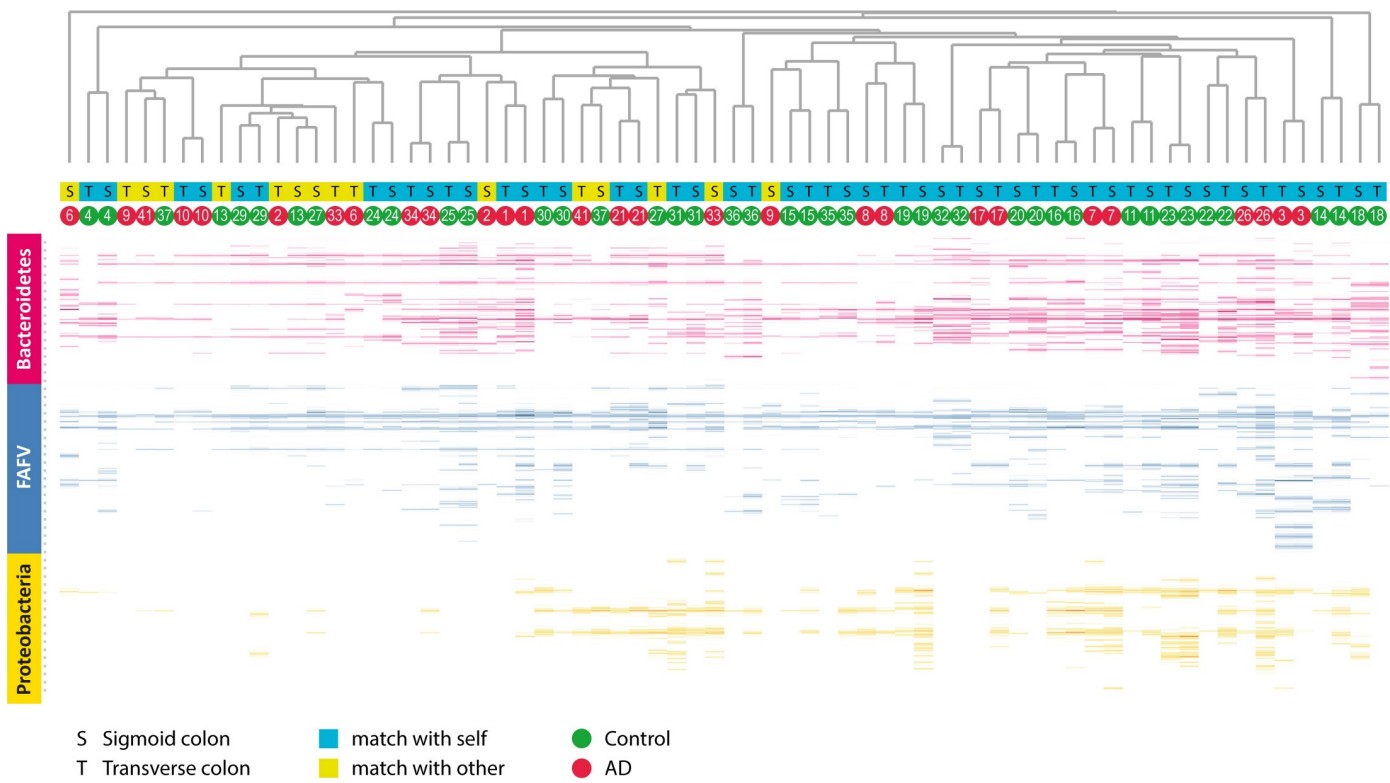

**Fig 4. Clustered heat map of all mucosal biopsy samples.** Each column represents a sample from either sigmoid (S) or transverse (T) colon. Each band represents a IS fragment corresponding to a specific bacterial species, the intensity to its abundance, the color to its phylum group. Generally, microbiota of the sigmoid colon shows highest correlation to the microbiota of the transverse colon of the same individual (all blue squares). Sometimes, correlation is stronger to a sample from another individual (yellow squares). However, no clear pattern can be seen here: samples from the asymptomatic diverticulosis (AD) group have a similar likelihood to having a low intra-individual correlation as control samples. Thus, random variation seems to be a more likely explanation here than an underlying biological phenomenon. Finally, there is no evident clustering of controls or AD individuals. FAFV: *Firmicutes/Actinobacteria/Fusobacteria/Verrucomicrobia*. The heat map was created with the Spotfire 10 software package (TIBCO, Palo Alto, CA, USA, https://www.tibco.com/products/tibco-spotfire).

containing a transverse colon and sigmoid colon biopsy of the same patient. This indicates that the intra-individual microbiota profiles showed a higher correlation than inter-individual microbiota profiles. Some biopsies clustered with a biopsy from another individual, but this was not related to the presence or absence of diverticula.

## Neutrophil and lymphocyte counts

On examination of the biopsies, none contained influx of neutrophils. Furthermore, we observed no difference in mean lymphocyte counts between both groups when we compared the lymphocyte counts, in the bottom of the crypts, or in the crypts as a whole. This was true for the sigmoid colon as well as the transverse colon (Table 2).

**Table 2. Mean lymphocyte counts.**

| Location | Asymptomatic diverticulosis group | Control group | P value |
|---|---|---|---|
| | N = 19 | N = 24 | |
| Transverse colon, bottom of crypt | 1.86 | 2.22 | 0,849 |
| Transverse colon, whole crypt | 9.56 | 8.39 | 0,261 |
| Sigmoid colon, bottom of crypt | 1.56 | 1.8 | 0,754 |
| Sigmoid colon, whole crypt | 6.95 | 7.12 | 0,765 |

## Discussion

In our study, intestinal microbiota profiles were highly similar within individuals for all phyla. Inter- individually, microbiota profiles differed substantially but without a clear association with the presence or absence of diverticula. Microbiota diversity in both sigmoid and transverse colon mucosal biopsies was comparable between both groups. We were not able to differentiate between the AD group and controls with a PLS-DA model based on the participants' microbiota composition or clustered heat map analysis. Mucosal lymphocyte counts were comparable between both groups. No neutrophils were detected in any of the studied biopsies. Our results suggest that compositional microbiota alterations and inflammation do not play a substantial role in the pathophysiology of diverticulosis. Our study did confirm the age-association with diverticulosis. The group of patients with diverticula was substantially older when compared to those without.

While AD by itself has no treatment currently, progression to symptomatic diverticulitis can lead to extensive surgery. Recently, there has been a paradigm shift pertaining to the etiology of diverticulitis. The traditional hypothesis that diverticulitis is caused by a fecalith obstruction of a diverticulum is replaced by the theory that diverticulitis has a multifactorial pathogenesis with an important role for the gut microbiota [9, 15]. We demonstrated previously that the gut microbiota in patients with diverticulitis differed from that of controls [9]. For diagnostic and possible treatment purposes, it would be interesting to see whether these changes in microbiota composition are also present in individuals before they develop diverticulitis. Moreover, if diverticulitis or development of diverticulosis could be predicted using microbiota analysis, this may result in more timely treatment and prevention of complications, and could allow for specific microbiota modulating therapies.

Few articles have been published on the gut microbiota composition in patients with AD. The largest study to date was conducted by Jones et al. [16]. They performed 16S-sequencing on mucosal samples of patients with AD and controls, and found weak associations between decreased relative abundance of *Proteobacteria* and *Comamonadaceae* and the presence of diverticula. They concluded that the mucosal adherent microbiota is unlikely to play a substantial role in development of diverticulosis, in accordance with our findings.

Tursi et al. used RT-PCR to detect specific microorganisms in fecal samples of patients with diverticulosis [17]. They reported no significant difference in total number of bacteria or in abundance of specific bacterial groups between patients with AD, patients with symptomatic uncomplicated diverticular disease (SUDD) and controls. Interestingly, *Akkermansia muciniphila* species was more abundant in patients with AD and SUDD than in controls. This is a remarkable finding, given *A. muciniphila* is considered a healthy gut commensal which is depleted in many inflammatory disorders [18]. Different to our study, fecal samples were analyzed using RT-PCR for microbiota composition, representing the luminal gut microbiota. This yields less information pertaining to the diverticular sites. Furthermore RT-PCR only allows for a limited number bacteria to be analyzed compared to 16S-based techniques.

Another study on microbiota alterations in diverticulosis was conducted by Barbara et al [6]. For microbiota analysis, 16S-sequencing was performed on both mucosal biopsies and stool samples. In concordance with our results, microbiota composition in colonic biopsies did not differ between patients with AD and controls. When AD and SUDD groups were combined and compared with controls, Barbara et al. observed a significant lower abundance of *Enterobacteriaceae* and a trend to higher abundance of *Bacteroides/Prevotella* in the combined AD/SUDD group. In feces, patients with SUDD had a significant decreased abundance of members of *Clostridium* cluster IX, *Fusobacteriae* and *Lactobacillaceae* compared to patients with AD.

Besides differences in microbiota composition, Barbara et al. focused on inflammatory markers in patients with diverticula. Identical to our results, they observed no difference in (T-)lymphocytes between patients with AD and patients without diverticula. Furthermore, two larger studies (n = 254 and n = 619) reported no association between colonic diverticula and numerous markers of mucosal and serological inflammation [19, 20]. In contrast, Barbara et al. observed a significantly higher macrophage count in patients with AD and SUDD compared with controls, suggesting there is some inflammatory process implicated in the pathophysiology of diverticulosis.

One of the strengths of our study is the use of mucosal biopsies for microbiota analysis instead of fecal samples. Because biopsies were collected from both the sigmoid and the transverse colon, we were able to assess potential local, diverticular related differences in microbiota composition. Another strength is that we used PLS-DA analysis, a sensitive technique in identifying biomarkers associated with disease state in complex data. Therefore, the lack of discriminating species found with PLS-DA does suggest the absence of a discriminative biomarker in this dataset.

Our study is hampered by several drawbacks. Firstly, our sample size is small and AD patients and controls were not matched by age. As AD is commonly found with increasing age, our study is not powered to find subtle, age-dependent differences in microbiota composition. Age is generally considered one of the main determinants of intestinal microbiota composition, however this begins to be relevant after the age of 70–75 years old, especially if no significant comorbidities or disability exist [21, 22]. This is in line with our finding that although the AD group was significantly older than the control group (66 vs 56 years respectively), the microbiota composition did not differ between the two groups. We only had limited availability to patient demographics. Several other factors such as diet can influence microbiota composition for which should be corrected [23].

Furthermore, using next-generation sequencing could reveal more subtle differences in microbiota composition in patients with- and without diverticula. 16S rRNA gene sequencing is currently seen as the gold standard for the profiling of the microbiome. However, for routine diagnostics or rapid processing of (small) sample batches, 16S rRNA gene sequencing is not well suited due to costs and time-consumption, creating space for other techniques. The IS-pro technique was developed with the goals of cost-effectiveness and simplicity in mind. Furthermore, analysis of proteomics and metabolomics could possibly reveal functional changes in the gut microbiota within an unchanged compositional microbiota. Comparison of the luminal microbiota (via fecal sample analysis) with mucosal biopsies could provide more insight in a possible interplay between these fractions of the gut microbiota. Another limitation is that microbiota analysis was performed on mucosal samples after bowel lavage, which has been shown to impact intestinal microbiota composition [24]. However, it is not possible to conduct a colonoscopy in a patient without prior bowel lavage. Another option might be to use rectal swabs or feces to verify whether the composition of the mucosal and fecal microbiota was consistent. Furthermore, collection of rectal swabs or feces is more practical and patient-friendly, although these methods have their own disadvantages as described before [24, 25]. Lastly, we only obtained lymphocyte and neutrophil counts as markers for inflammation. An assessment of other immune cells such as macrophages, or cytokines in our data set could provide a more detailed view of the mucosal immunity in patients with AD and should be evaluated in future studies.

In conclusion, microbiota composition and inflammatory markers were comparable among patients with asymptomatic diverticulosis and controls. This suggests that the gut microbiota composition and mucosal inflammation do not play a major role in the pathogenesis of diverticula formation. Whether microbiota and mucosal inflammatory changes as have

been observed in diverticulitis are cause or sequelae of the disease remains unclear and merits further investigation.

## Supporting information

**S1 File. Patient characteristics.**
(XLSX)

**S2 File. Microbiota data.**
(XLSX)

## Author Contributions

**Conceptualization:** Johan Ph. Kuyvenhoven, René Buijsman, Marja A. Boermeester, Hein B. A. C. Stockmann, Niels de Korte.

**Data curation:** Johan Ph. Kuyvenhoven, René Buijsman, Marja A. Boermeester, Niels de Korte.

**Formal analysis:** Johan Ph. Kuyvenhoven, Anat Eck, Herman Bril, Niels de Korte, Andries E. Budding.

**Investigation:** Rogier E. Ooijevaar, Johan Ph. Kuyvenhoven, Herman Bril, René Buijsman, Hein B. A. C. Stockmann, Niels de Korte, Andries E. Budding.

**Methodology:** Tessel M. van Rossen, Johan Ph. Kuyvenhoven, Anat Eck, Herman Bril, René Buijsman, Marja A. Boermeester, Hein B. A. C. Stockmann, Niels de Korte, Andries E. Budding.

**Software:** Anat Eck, Andries E. Budding.

**Supervision:** Johan Ph. Kuyvenhoven, Marja A. Boermeester, Hein B. A. C. Stockmann.

**Validation:** Anat Eck.

**Visualization:** Tessel M. van Rossen, Andries E. Budding.

**Writing – original draft:** Tessel M. van Rossen, Rogier E. Ooijevaar, Niels de Korte, Andries E. Budding.

**Writing – review & editing:** Tessel M. van Rossen, Rogier E. Ooijevaar, Johan Ph. Kuyvenhoven, Anat Eck, Herman Bril, René Buijsman, Marja A. Boermeester, Hein B. A. C. Stockmann, Niels de Korte, Andries E. Budding.

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
