## [Decision Letter · Decision Letter 0]

20 May 2021

PONE-D-21-08956

Microbiota composition and mucosal immunity in patients with asymptomatic diverticulosis and controls

PLOS ONE

Dear Dr. van Rossen,

Thank you for submitting your manuscript to PLOS ONE. After careful consideration, we feel that it has merit but does not fully meet PLOS ONE’s publication criteria as it currently stands. Therefore, we invite you to submit a revised version of the manuscript that addresses the points raised during the review process.

Specific concerns related to patient inclusion criteria and available metadata to contextualize the results are required to explain the novelty and reliability of the current analysis.

We look forward to receiving your revised manuscript.

Kind regards,

Christopher Staley, Ph.D.

Academic Editor

PLOS ONE

Additional Editor Comments:

The study addresses an important and emerging question concerning the role of the intestinal microbiota in diverticular disease. As both reviewers acknowledge, age is an important confounding factor, in addition to other exclusion requirements ie, antibiotic exposure or other medications and clinical metadata including immunological assays or dietary surveys are needed to properly contextualize these data. Without these, this is a well conducted survey but provides little novel information.

Journal Requirements:

2. We note that you are reporting an analysis of a microarray, next-generation sequencing, or deep sequencing data set. PLOS requires that authors comply with field-specific standards for preparation, recording, and deposition of data in repositories appropriate to their field. Please upload these data to a stable, public repository (such as ArrayExpress, Gene Expression Omnibus (GEO), DNA Data Bank of Japan (DDBJ), NCBI GenBank, NCBI Sequence Read Archive, or EMBL Nucleotide Sequence Database (ENA)). In your revised cover letter, please provide the relevant accession numbers that may be used to access these data. For a full list of recommended repositories, see http://journals.plos.org/plosone/s/data-availability#loc-omics or http://journals.plos.org/plosone/s/data-availability#loc-sequencing.

"I have read the journal's policy and the authors of this manuscript have the following competing interests: AEB reports personal fees of inBiome B.V. (employee/stock holder)."

We note that one or more of the authors are employed by a commercial company: inBiome B.V..

4.1. Please provide an amended Funding Statement declaring this commercial affiliation, as well as a statement regarding the Role of Funders in your study. If the funding organization did not play a role in the study design, data collection and analysis, decision to publish, or preparation of the manuscript and only provided financial support in the form of authors' salaries and/or research materials, please review your statements relating to the author contributions, and ensure you have specifically and accurately indicated the role(s) that these authors had in your study. You can update author roles in the Author Contributions section of the online submission form.

4.2. Please also provide an updated Competing Interests Statement declaring this commercial affiliation along with any other relevant declarations relating to employment, consultancy, patents, products in development, or marketed products, etc.  

5. Thank you for submitting the above manuscript to PLOS ONE. During our internal evaluation of the manuscript, we found significant text overlap between your submission and the following previously published works, some of which you are an author.

https://research.vu.nl/en/publications/diverticulitis-insights-in-aetiology-and-treatment

Please revise the manuscript to rephrase the duplicated text, cite your sources, and provide details as to how the current manuscript advances on previous work. Please note that further consideration is dependent on the submission of a manuscript that addresses these concerns about the overlap in text with published work.

Reviewers' comments:

Reviewer's Responses to Questions

**Comments to the Author**

1. Is the manuscript technically sound, and do the data support the conclusions?

Reviewer #1: Yes

Reviewer #2: No

2. Has the statistical analysis been performed appropriately and rigorously? 

Reviewer #1: Yes

Reviewer #2: Yes

3. Have the authors made all data underlying the findings in their manuscript fully available?

Reviewer #1: Yes

Reviewer #2: Yes

4. Is the manuscript presented in an intelligible fashion and written in standard English?

Reviewer #1: Yes

Reviewer #2: Yes

5. Review Comments to the Author

Reviewer #1: Excellent manuscript showing that the presence of asymptomatic diverticulosis is not associated with significant alterations of intestinal mucosal microbiota. The findings suggest that gut microbiota should not have a major role in the pathogenesis of diverticula. The topic is not completely novel, since there are already some studies investigating the same research question (correctly acknowledged in the discussion). However, the present study has been conducted on a larger sample size and with a sounder methodology than its counterparts.

I have just three comments/suggestions:

1) Cases (patients with diverticulosis) and controls were not matched by age. This is an important limitation of the study. In fact, age is generally considered one of the main determinants of intestinal microbiota composition. However, the effect of age on microbiota begins to be relevant after the age of 70-75 years old, especially is no significant comorbidities or disability exist. Thus, the effect of different age on the microbiota composition of the two groups enrolled in this study could have been negligible, provided that the burden of comorbidities and disability of the two groups was similar. I suggest that the authors include a comment on these issues in their manuscript.

2) No prespecified sample size calculation seems to have been performed for this study. Please specify that the study was conducted on a convenience sample size.

3) It would have been interesting to include in the study also an analysis of the fecal microbiota of participants, to verify whether the composition of the mucosal and fecal microbiota was consistent. Fecal microbiota is much more accessible for analyses, also from a clinical perspective.

Reviewer #2: The topic of this manuscript is very interesting. The pathophysiology of diverticular disease is still incompletely known and the role of microbiota and immunity is only partially reported in the literature. This study evaluated mucosal microbiota and immunity in asymptomatic diverticulosis compared with a control group without diverticula. there are, however, several limitations that make it difficult to support the conclusions speculated by the authors and make the results unreliable. among the exclusion criteria, the authors did not consider the use of antibiotics and laxatives which are factors changing gut microbiota. the lack of information about diet generates doubts about the results. Asymptomatic diverticulosis and controls significantly differ for age, which again influences gut microbiota. without correcting data for this variable, there is likely a bias in the results. Concerning immunity, the study did not look at macrophages, which as authors themselves reported in the manuscript, have been found to be different in diverticular disease compared to controls. all these limitations weaken the results and make it difficult to support the conclusions of the authors.

6. PLOS authors have the option to publish the peer review history of their article (what does this mean?). If published, this will include your full peer review and any attached files.

Reviewer #1: No

Reviewer #2: No

---

## [Author Response · Author response to Decision Letter 0]

14 Jul 2021

Amsterdam, 02-07-2021

Dear Mr Staley,

Thank you for the review of our manuscript entitled “Microbiota composition and mucosal inflammation in patients with asymptomatic diverticulosis and controls” (PONE-D-21-08956). With these valuable comments we were able to improve our manuscript.

Below I copied the reviewers’ comments and our response to each comment.

We look forward to your response and hope that you find our revised manuscript eligible for publication in PLOS ONE.

Sincerely,

Tess van Rossen

PhD candidate, department of Medical Microbiology & Infection Control, Amsterdam UMC

Reviewer comments:

Reviewer #1: 

Excellent manuscript showing that the presence of asymptomatic diverticulosis is not associated with significant alterations of intestinal mucosal microbiota. The findings suggest that gut microbiota should not have a major role in the pathogenesis of diverticula. The topic is not completely novel, since there are already some studies investigating the same research question (correctly acknowledged in the discussion). However, the present study has been conducted on a larger sample size and with a sounder methodology than its counterparts.

I have just three comments/suggestions:

1) Cases (patients with diverticulosis) and controls were not matched by age. This is an important limitation of the study. In fact, age is generally considered one of the main determinants of intestinal microbiota composition. However, the effect of age on microbiota begins to be relevant after the age of 70-75 years old, especially if no significant comorbidities or disability exist. Thus, the effect of different age on the microbiota composition of the two groups enrolled in this study could have been negligible, provided that the burden of comorbidities and disability of the two groups was similar. I suggest that the authors include a comment on these issues in their manuscript.

Thank you for the review of our manuscript and for your positive response. We agree that this is a limitation of the study. As you suggested, we included a comment and corresponding references in the discussion section (lines 327-333):

“Our study is hampered by several drawbacks. Firstly, our sample size is small and AD patients and controls were not matched by age. As AD is commonly found with increasing age, our study is not powered to find subtle, age-dependent differences in microbiota composition. Age is generally considered one of the main determinants of intestinal microbiota composition; however this begins to be relevant after the age of 70-75 years old, especially if no significant comorbidities or disability exist (21,22). This is in line with our finding that although the AD group was significantly older than the control group (66 vs 56 years respectively), the microbiota composition did not differ between the two groups.”

In addition, we re-assessed electronic patient records of included patients to evaluate potential differences in comorbidities and disability of the two groups. We found that the burden of disability was similar in the two groups. The results are displayed in Table 1 and included in the Results section (lines 175-179):

“The number of patients with specific comorbidities was similar in the two groups, except that gastro-intestinal disorders and malignancies (but not colon cancer) were more frequent in the control group compared to the AD group. Gastro-intestinal comorbidity in the control group concerned esophagitis (n=2), aspecific colitis (n=1) and irritable bowel syndrome (n=1).”

2) No prespecified sample size calculation seems to have been performed for this study. Please specify that the study was conducted on a convenience sample size.

Thank you for this suggestion, we included this statement in the Methods section (lines 87-88).

3) It would have been interesting to include in the study also an analysis of the fecal microbiota of participants, to verify whether the composition of the mucosal and fecal microbiota was consistent. Fecal microbiota is much more accessible for analyses, also from a clinical perspective.

We agree that an additional analysis of fecal samples (or rectal swabs, also a practical sample type from a clinical perspective) would have been very interesting for comparison between the luminal and fecal microbiota composition. Unfortunately, feces was not collected in this study but we included this suggestion/limitation in the Discussion section (lines 352-356):

“Another option might be to use rectal swabs or feces to verify whether the composition of the mucosal and fecal microbiota was consistent. Furthermore, collection of rectal swabs or feces is more practical and patient-friendly, although these methods have their own disadvantages as described before (24, 25).”

Reviewer #2: 

The topic of this manuscript is very interesting. The pathophysiology of diverticular disease is still incompletely known and the role of microbiota and immunity is only partially reported in the literature. This study evaluated mucosal microbiota and immunity in asymptomatic diverticulosis compared with a control group without diverticula. There are, however, several limitations that make it difficult to support the conclusions speculated by the authors and make the results unreliable. Among the exclusion criteria, the authors did not consider the use of antibiotics and laxatives which are factors changing gut microbiota. The lack of information about diet generates doubts about the results. 

Thank you for the review of our manuscript your valuable suggestions.

Concerning the exclusion criteria, patients with antibiotics and laxative use were indeed not excluded beforehand. However, we re-assessed electronic patient records of our study population and found that none of the patients received antibiotics, and only few patients used laxatives (AD group: 3 patients, control group: 1 patient). This data is added to the text of the Results section and Table 1 (page 9). Furthermore, all patients underwent routine bowel lavage (high dose laxatives) for colonoscopy so we assume that the potential effect of prior laxative use on the mucosal microbiota composition is probably marginal compared to the effect of the bowel preparation. Unfortunately, data on diet was lacking, since this factor is not regularly reported in patient records. We addressed this limitation in the Discussion section (lines 334-336):

“We only had limited availability to patient demographics. Several other factors such as diet can influence microbiota composition for which should be corrected (23)."

Asymptomatic diverticulosis and controls significantly differ for age, which again influences gut microbiota. Without correcting data for this variable, there is likely a bias in the results. 

Thank you for this comment. See also the answer on comment 1 of reviewer #1: We agree that this is a limitation of the study. However, since the microbiota composition of the AD group and control group was comparable despite the difference in age, we do not expect that age has influenced the results. Furthermore, previous studies show that the effect of age on the microbiota composition begins to be relevant after the age of 70-75 years old, while the age of the AD patients and controls was 66 and 56 years old, respectively. Nevertheless, we included this limitation and corresponding references in the discussion section (lines 327-333):

“Our study is hampered by several drawbacks. Firstly, our sample size is small and AD patients and controls were not matched by age. As AD is commonly found with increasing age, our study is not powered to find subtle, age-dependent differences in microbiota composition. Age is generally considered one of the main determinants of intestinal microbiota composition; however this begins to be relevant after the age of 70-75 years old, especially if no significant comorbidities or disability exist (21,22). This is in line with our finding that although the AD group was significantly older than the control group (66 vs 56 years respectively), the microbiota composition did not differ between the two groups.”

Concerning immunity, the study did not look at macrophages, which as authors themselves reported in the manuscript, have been found to be different in diverticular disease compared to controls. 

Thank you for this comment. We agree that analysis of macrophages would have been an interesting addition to our study. Unfortunately, this data was not collected in our study set, but we addressed this limitation and suggestion for future studies in the Discussion section (lines 356-358):

“An assessment of other immune cells such as macrophages, or cytokines in our data set could provide a more detailed view of the mucosal immunity in patients with AD and should be evaluated in future studies.”

Additional Editor Comments:

The study addresses an important and emerging question concerning the role of the intestinal microbiota in diverticular disease. As both reviewers acknowledge, age is an important confounding factor, in addition to other exclusion requirements ie, antibiotic exposure or other medications and clinical metadata including immunological assays or dietary surveys are needed to properly contextualize these data. Without these, this is a well conducted survey but provides little novel information.

Thank you for the review of our manuscript and the positive feedback. We agree that the metadata is important for correct interpretation of the results on the role of the intestinal microbiota in diverticulosis and diverticulitis. Therefore, we re-assessed electronic patient records to extract data on potential confounders, such as smoking, alcohol use, antibiotic exposure, and comorbidities. We found no large differences between the two groups. The results are displayed in Table 1 and included in the Results section (lines 173-179):

“Six out of 19 AD patients were smokers, compared to 3/24 controls, and alcohol consumption was slightly higher in the AD group compared to the control group. None of the patients received antibiotics via the hospital prior to colonoscopy. The number of patients with specific comorbidities was similar in the two groups, except that gastro-intestinal disorders and malignancies (but not colon cancer) were more frequent in the control group compared to the AD group. Gastro-intestinal comorbidity in the control group concerned esophagitis (n=2), aspecific colitis (n=1) and irritable bowel syndrome (n=1).”

---

## [Decision Letter · Decision Letter 1]

12 Aug 2021

Microbiota composition and mucosal immunity in patients with asymptomatic diverticulosis and controls

PONE-D-21-08956R1

Dear Dr. van Rossen,

We’re pleased to inform you that your manuscript has been judged scientifically suitable for publication and will be formally accepted for publication once it meets all outstanding technical requirements.

Kind regards,

Christopher Staley, Ph.D.

Academic Editor

PLOS ONE

Additional Editor Comments (optional):

Reviewers' comments:

Reviewer's Responses to Questions

**Comments to the Author**

1. If the authors have adequately addressed your comments raised in a previous round of review and you feel that this manuscript is now acceptable for publication, you may indicate that here to bypass the “Comments to the Author” section, enter your conflict of interest statement in the “Confidential to Editor” section, and submit your "Accept" recommendation.

Reviewer #1: All comments have been addressed

Reviewer #2: All comments have been addressed

2. Is the manuscript technically sound, and do the data support the conclusions?

Reviewer #1: Yes

Reviewer #2: Yes

3. Has the statistical analysis been performed appropriately and rigorously? 

Reviewer #1: Yes

Reviewer #2: Yes

4. Have the authors made all data underlying the findings in their manuscript fully available?

Reviewer #1: Yes

Reviewer #2: Yes

5. Is the manuscript presented in an intelligible fashion and written in standard English?

Reviewer #1: Yes

Reviewer #2: Yes

6. Review Comments to the Author

Reviewer #1: The authors have adequately responded to all my previous comments and correctly acknowledged the study limitations. I have no further concerns.

Reviewer #2: (No Response)

7. PLOS authors have the option to publish the peer review history of their article (what does this mean?). If published, this will include your full peer review and any attached files.

Reviewer #1: No

Reviewer #2: No

---

## [Editor Report · Acceptance letter]

27 Aug 2021

PONE-D-21-08956R1 

Microbiota composition and mucosal immunity in patients with asymptomatic diverticulosis and controls 

Dear Dr. van Rossen:

I'm pleased to inform you that your manuscript has been deemed suitable for publication in PLOS ONE. Congratulations! Your manuscript is now with our production department. 

Kind regards, 

on behalf of

Dr. Christopher Staley 

Academic Editor

PLOS ONE